# Comparative Effectiveness of Kyphoplasty and Radiation with or Without Radiofrequency Ablation in Spinal Metastases from Lung Cancer

**DOI:** 10.3390/healthcare13233101

**Published:** 2025-11-28

**Authors:** Kamal Shaik, Spencer T. Rasmussen, Muhammad Hozien, Rudy Rahme, Michael Karsy

**Affiliations:** 1Department of Neurosurgery, Drexel University College of Medicine, Philadelphia, PA 19104, USA; kas685@drexel.edu (K.S.); str62@drexel.edu (S.T.R.); mnh86@drexel.edu (M.H.); 2Global Neurosciences Institute, Atlantic City, NJ 08401, USA; rr999@drexel.edu; 3Department of Neurosurgery, University of Michigan Medical School, Ann Arbor, MI 48109, USA

**Keywords:** spinal metastases, lung cancer, kyphoplasty, radiofrequency ablation, radiotherapy, TriNetX

## Abstract

**Highlights:**

**What are the main findings?**

Adjunctive radiofrequency ablation (RFA) with kyphoplasty and 
radiotherapy did not significantly improve survival, recurrence, neurologic, or pain outcomes compared to kyphoplasty and radiotherapy alone.Propensity-matched analysis of over 700 patients demonstrated 
equivalent effectiveness between treatment groups.

**What is the implication of the main finding?**

The incremental benefit of adding RFA to multimodal management of 
lung cancer spinal metastases appears limited.Future prospective studies are needed to identify subgroups that may derive clinical advantage from RFA.

**Abstract:**

**Background/Objectives:** Spinal metastases from lung cancer cause substantial pain, instability, and neurologic compromise. Radiotherapy and kyphoplasty are standard treatment modalities, while radiofrequency ablation (RFA) has emerged as a potential adjunct for cytoreduction. The objective of this study was to evaluate whether RFA confers additional benefit when combined with kyphoplasty and radiotherapy in patients with lung cancer spinal metastases. **Methods:** We conducted a retrospective cohort study of adults with lung cancer and spinal metastases from 2012–2024 using the TriNetX database. Patients were identified using International Classification of Disease, Tenth Revision, Clinical Modification (ICD-10-CM) codes and stratified into two groups: kyphoplasty with radiotherapy alone versus kyphoplasty with radiotherapy and RFA. Propensity score matching was applied to balance demographic and clinical covariates. The primary outcome was 1-year all-cause mortality. Secondary outcomes included tumor recurrence, neurologic complications, and pain burden as assessed by opioid prescription rates. Risk ratios (RR) with 95% confidence intervals (CI) were calculated. **Results:** A total of 703 patients met inclusion criteria. After matching, no significant differences were observed between groups for 1-year mortality (RR 1.021, 95% CI 0.83–1.256), tumor recurrence (RR 0.989, 95% CI 0.789–1.238), neurologic complications (RR 1.052, 95% CI 0.563–1.967), or opioid use as a pain proxy (RR 0.986, 95% CI 0.76–1.28). **Conclusions:** The addition of RFA to kyphoplasty and radiotherapy did not significantly impact survival, recurrence, neurologic, or pain outcomes in patients with spinal metastases from lung cancer. These findings suggest that the incremental benefit of RFA in this setting is limited and emphasize the need for prospective studies to refine patient selection and treatment strategies.

## 1. Introduction

Spinal metastases are a common and clinically significant manifestation of advanced malignancy, occurring in many cancer patients during the disease course [1]. Lung cancer is among the most common primary tumors to metastasize to the vertebrae, significantly contributing to morbidity through pain, vertebral instability, pathological fractures, and neurological impairment [2,3].

Management of vertebral metastases typically involves a multimodal approach, specifically radiotherapy and surgery [4]. Radiotherapy remains a mainstay of treatment for symptomatic spinal lesions and provides palliative tumor control and analgesia [5,6,7,8]. Concurrently, vertebral augmentation techniques such as kyphoplasty offer mechanical stabilization, reduce micro-motion pain, and can rapidly improve symptoms in patients with pathologic fractures or impending collapse [9]. In metastatic lung cancer specifically, balloon kyphoplasty has been shown to correct kyphotic deformities, improve quality of life, and safely palliate pain. However, it has not been demonstrated to prolong survival [10].

Over the past decade, radiofrequency ablation (RFA) has gained traction as a minimally invasive adjunct in the management of vertebral metastases [11,12,13]. RFA delivers localized thermal energy to induce tumor necrosis and reduce viable tumor burden [14]. Thus, by cytoreducing tumor volume, RFA is posited to potentiate the efficacy of radiotherapy by reducing radioresistant tumor burden, decrease pain via nerve fiber ablation, and create a more favorable environment for vertebral stabilization with less risk of cement leakage [15].

Several clinical studies suggest that combining RFA with vertebral augmentation or radiotherapy is both feasible and generally safe, yielding meaningful pain relief and local control in many patients. In a recent systematic review, most studies reported highly effective pain control, with low rates of serious complications [16,17]. In another meta-analysis of studies combining RFA and cement augmentation, favorable outcomes in pain, function, and safety were reported across heterogenous tumor types, though the authors cautioned that long-term data remain limited [18]. Furthermore, studies using navigational bipolar RFA also report high rates of pain relief and radiographic local control with acceptable safety profiles [19].

Despite these encouraging findings, the incremental benefit of RFA in addition to kyphoplasty and radiotherapy is far from established. Many existing reports are small, single-institution series with sparse evidence comparing outcomes such as overall survival, tumor recurrence, neurological complications, or pain trajectories between patients treated with kyphoplasty and radiotherapy versus those receiving kyphoplasty, radiotherapy, and RFA [20,21,22,23]. Given this gap, a comparative effectiveness analysis leveraging a large real-world database offers an opportunity to clarify the role of RFA in lung cancer spinal metastases. In this study using the TriNetX database, we compare outcomes between patients undergoing kyphoplasty with radiotherapy alone versus those receiving adjunct RFA. Our goal was to determine whether RFA confers clinically meaningful advantages in survival, recurrence, neurologic outcomes, or pain burden in this specific population.

## 2. Materials and Methods

### 2.1. Data Source

The retrospective analysis utilized the TriNetX Research Network, a global, federated health research platform that aggregates the de-identified electronic medical record data from participating healthcare organizations (HCO). These HCOs include academic medical centers, specialty physician networks, and integrated delivery systems across the United States and internationally [24]. The specific entities and locations of contributing HCOs cannot be disclosed due to contractual and privacy agreement.

Each participation institution provides structured patient level clinical data that are extracted directly from electronic health record systems (EHR), laboratory systems, and pharmacy systems. These data are then normalized, mapped, and curated by TriNetX to standard terminologies including International Classification of Disease, Tenth Revision, Clinical Modification (ICD-10-CM) for diagnoses, Current Procedural Terminology (CPT) and International Classification of Diseases, 10th Revision, Procedure Coding System (ICD-10-PCS) for procedures, RxNorm for medications, and Logical Observation Identifiers Names and Codes (LOINC) for laboratory values. All data are de-identified in accordance with the Health Insurance Portability and Accountability Act (HIPAA) Privacy Rule (45 CFR §164.514(a)) through expert determination [24].

The resulting dataset includes longitudinal information in demographics, diagnoses, procedures, medications, laboratory results, and encounter details representing over 170 million unique patient records. Contributing institutions update their data at regular intervals, ensuring that analyses reflect near real-time information derived from clinical practice [24]. However, neither patient-level identifiers nor HCO-level information are available to users or disclosed externally.

### 2.2. Cohort Selection

All patients ≥ 18 years of age diagnosed with lung cancer metastases to the spine between January 2012 and December 2024 were included in this retrospective cohort study. ICD-10-CM codes were used for inclusion and exclusion criteria for patient cohorts. Eligible diagnoses required evidence of malignant neoplasm of the bronchus and lung (ICD-10-CM C34) in combination with secondary malignant neoplasm of bone (ICD-10-CM C79.51). Patients with malignant neoplasms of the vertebral column (ICD-10-CM C41.2); skull and face bones (ICD-10-CM C41.0); or pelvic bones, sacrum, and coccyx (ICD-10-CM C41.4) were excluded to avoid the inclusion of primary cancers of the bone.

Procedural eligibility was defined by the presence of percutaneous vertebral augmentation, identified through the following CPT codes: thoracic vertebral augmentation (CPT 22513), lumbar vertebral augmentation (CPT 22514), and each additional thoracic or lumbar level (CPT 22515). Patients undergoing ablation were identified by CPT 20982, which captures percutaneous radiofrequency ablation of one or more bone tumors with imaging guidance. Radiotherapy was identified using the TriNetX curated radiation exposure code (TNX Curated 1081).

Based on procedural exposure, patients were stratified into two cohorts: (1) kyphoplasty with radiotherapy and RFA, and (2) kyphoplasty with radiotherapy without RFA, as shown in Figure 1. This grouping allowed direct comparison of outcomes between multimodal management with and without adjunctive ablation.

### 2.3. Variables

All covariates used for matching were assessed on or before the day preceding the defined index event (kyphoplasty and radiotherapy with or without RFA). This ensures that only variables present prior to treatment were included, thus maintaining the correct temporal ordering for covariate evaluation. The TriNetX platform applies temporal logic to analytic queries by automatically restricting covariates to values observed prior to the exposure of interest. While the system does not currently support user-defined lag periods between covariate measurement and the exposure date, this pre-index alignment mitigates potential reverse causation or immortal time bias by excluding post-treatment variables from the propensity model. The absence of explicit lag-adjusted windows is a limitation of the platform but is partially offset by the conservative covariate timing logic. Additionally, covariates were selected based on their clinical plausibility as confounders and included only if reliably documented in structured EHR data. This approach ensures that all baseline variables were known to the clinical team prior to the index event and reflects common real-world observational practices.

Demographic factors included age at index, sex (male or female), and race/ethnicity categorized as White, Black or African American, Asian, Hispanic or Latino, and Native Hawaiian or Pacific Islander. Clinical comorbidities incorporated into the model included the following: heart failure (ICD-10-CM I50), type 2 diabetes mellitus (ICD-10-CM E11), chronic kidney disease (ICD-10-CM N18), and ischemic heart disease (ICD-10-CM I25). Treatment-related covariates included prior chemotherapy exposure (TNX Curated 10301) and prior encounters for antineoplastic radiation therapy (ICD-10-CM Z51.0)

### 2.4. Outcome Measures

The primary outcome of interest was all-cause mortality within one year of the index event. Secondary outcomes included tumor recurrence, neurologic complications, and pain burden. Tumor recurrence was defined as subsequent documentation of secondary malignant neoplasm of bone (ICD-10-CM C79.51) or repeat coding of subsequent vertebral augmentation procedures (CPT 22513, 22514, 22515) occurring after the index event. Neurologic complications were identified using ICD-10-CM codes for spinal cord compression (G95.20, G95.29), cauda equina syndrome (G83.4), other and unspecified disease of the spinal cord (G95), and other disorders of the central nervous system (G96). Pain burden was evaluated indirectly using opioid prescriptions as a surrogate from severe pain, captured from TriNetX medication records. Specifically, prescriptions for morphine (RxNorm 7052), oxycodone (RxNorm 7804), hydromorphone (RxNorm 3423), fentanyl (RxNorm 4337), tramadol (RxNorm 10689), and codeine (RxNorm 2670) were used as indicators of opioid utilization. All outcomes were followed over a one-year period.

### 2.5. Statistical Analysis

Comparative risk analyses were performed using the TriNetX Compare Outcomes analytic card, which calculates risk ratios (RR) and corresponding 95% confidence intervals (CI) for binary outcomes between cohorts. Risk was defined as the proportion of patients within each cohort who experienced the outcome of interest (e.g., mortality, tumor recurrence, opioid utilization) during a fixed one-year follow-up window starting on the day after the index event.

The risk ratio was calculated by dividing the risk in the exposed cohort (kyphoplasty with radiotherapy and RFA) by the risk in the unexposed or comparator cohort (kyphoplasty with radiotherapy without RFA), meaning that the cohort receiving kyphoplasty with radiotherapy without RFA served as the reference group for RR calculation. RRs > 1 indicate a higher risk in the exposed group, RRs < 1 indicate a lower risk, and RRs = 1 indicate no difference. The 95% confidence intervals for RRs are computed using the Wald method, based on a log-binomial approximation. Specifically, the natural logarithm of the RR is calculated, and its standard error is used to construct the CI bounds, which are then exponentiated to yield the final interval on the risk ratio scale. *p*-values are generated using a two-sided test of the log risk ratio. These calculations are automated within the TriNetX platform and validated across multiple studies, but the platform does not expose patient-level data or intermediate tables due to privacy constraints. The analytic assumes independent binary outcomes and does not support additional covariate adjustment beyond what is achieved through propensity score matching.

Propensity score matching was implemented to minimize baseline differences and reduce selection bias. TriNetX employs a greedy nearest-neighbor algorithm with a 1:1 matching ratio based on selected covariates. Propensity scores were generated within the TriNetX platform using logistic regression modeling. This model estimates the probability of receiving RFA as part of the treatment pathway based on the covariates selected a priori. All demographic, clinical, and treatment-related variables listed in Section 2.3—including age, sex, race/ethnicity, comorbidities (heart failure, type 2 diabetes mellitus, chronic kidney disease, ischemic heart disease), prior chemotherapy, and prior radiation—were included as predictors in the model. The platform’s logistic regression algorithm outputs a scalar propensity score for each patient representing the conditional probability of treatment assignment, which was then used to perform 1:1 greedy nearest-neighbor matching without replacement. This matching approach minimizes the absolute difference in propensity scores between treated and comparator patients. The internal coefficients and diagnostics of the logistic model are not exposed within the TriNetX interface due to data protection standards and de-identification protocols.

Covariate balance following propensity score matching was evaluated using standard mean differences (SMD), which are the recommended method for assessing comparability between matched groups [25]. SMDs quantify the absolute difference in means or proportions between cohorts, standardized by the pooled standard deviation. Unlike *p*-values, SMDs are independent of sample size and directly measure the magnitude of difference. A threshold of <0.1 is commonly used to indicate acceptable balance for each covariate [25]. In this study, all demographic and clinical covariates included in the matching model had post-match SMDs below this threshold, as shown in Table 1, confirming successful balance between the treatment groups. The TriNetX platform does not currently support additional multivariable regression adjustment post-matching; however, given the adequate balance demonstrated by SMDs, residual confounding is unlikely to significantly bias the results.

## 3. Results

A total of 703 patients met the inclusion criteria, comprising 202 patients who underwent kyphoplasty with radiotherapy and RFA, and 501 patients who only underwent kyphoplasty with radiotherapy. Following propensity score matching, baseline demographic and clinical characteristics were similar between the patient groups, as shown in Table 1.

Patients who underwent kyphoplasty and radiotherapy with adjunctive RFA demonstrated no significant differences in outcomes compared to those treated with kyphoplasty and radiotherapy alone (Table 2, Figure 2). Specifically, the addition of RFA was not associated with altered risk of one-year mortality (RR: 1.021; 95% CI: 0.83–1.256; *p* = 0.8420), neurologic complications (RR: 1.052; 95% CI: 0.563–1.967; *p* = 0.8732), tumor recurrence (RR: 0.989; 95% CI: 0.789–1.238; *p* = 0.9199), or pain burden assessed via opioid prescriptions (RR: 0.986; 95% CI: 0.76–1.28; *p* = 0.9174).

## 4. Discussion

In this large, real-world cohort study of patients with lung cancer metastatic to the spine, we compared outcomes following kyphoplasty and radiotherapy with and without RFA. After propensity score matching, our analysis demonstrated no statistically significant differences between groups in one-year all-cause mortality, tumor recurrence, neurologic complications, or pain burden assessed via opioid prescriptions at 1 year follow-up. Clinically, these findings suggest that the addition of RFA to kyphoplasty and radiotherapy does not confer incremental clinical benefit in this patient population. These results suggests that the main value of RFA may lie in symptom-specific, case-by-case use, rather than broad routine application. Ideal subgroups of patients benefiting from RFA require further characterization.

### 4.1. Mortality

The null association between RFA addition and one-year mortality reinforces the concept that RFA is not likely to alter overall survival in the setting of metastatic lung cancer to the spine. This is consistent with expectations, as systemic disease burden and response to systemic therapies typically drive survival in this population rather than local interventions.

Our results are similar to prior studies. Tomasian et al. described percutaneous thermal ablation of spinal metastases as safe and effective in symptom control, but did not report definitive survival benefits [26]. Similarly, Giammalva et al. described combined RFA with vertebroplasty in metastatic spinal fractures and noted improvements in pain and function but did not attribute survival prolongation to the procedure [27]. Additional studies suggest that RFA should be viewed as a palliative, symptom-directed adjunct [28]. Had RFA had a substantial effect on local disease burden, one might expect secondary effects on morbidity or local failure that could influence survival over time. Nonetheless, our results were unable to detect this effect. One possibility may be the need for longer follow-up than 1 year or consideration of other adjuvant radiotherapy and chemotherapy treatments for this patient population to help discern subgroups of survival benefit. This may be an area for future study.

### 4.2. Tumor Recurrence

We did not identify significant reduction in tumor recurrence with the addition of RFA in the metastatic lung cancer population. There may be several reasons for these findings. The ablation zones created by RFA may simply be insufficient to impact outcome or miss microscopic disease [29]. Moreover, it is plausible that recurrence beyond the vertebral segment (e.g., adjacent levels or distant metastases) would not be prevented by local ablation, further diluting any observed effect on overall recurrence metrics [30]. Lastly, the involvement of modern stereotactic radiosurgery and body radiotherapy may be sufficient for disease control.

Published literature demonstrates favorable local control rates after RFA in vertebral metastases, however these results are limited by inclusion of mixed tumor types [31]. Mayer et al. reported high rates of local control and favorable safety outcomes for bipolar RFA [19]. Ragheb et al. found that combining RFA with vertebral cement augmentation was associated with reduced local tumor recurrence from metastatic breast cancer and comparable pain relief when compared to kyphoplasty alone [22]. Nonetheless, many of these series are uncontrolled, of short duration, and with heterogeneous patient populations, which may overestimate the real-world incremental benefit [32]. Our comparative design provides a larger, multi-center, real-world evaluation of the additional value for RFA with kyphoplasty and radiotherapy; in our data, it did not demonstrably reduce the hazard of recurrence within one year.

### 4.3. Neurologic Complications

We did not detect improved or worsening neurologic complications with the addition of RFA. This is reassuring from a safety perspective, given theoretical concerns about thermal injury to spinal cord or nerve roots.

In support of procedural safety, multiple studies of percutaneous spinal ablation report low rates of neurologic sequelae. Tomasian et al. characterized percutaneous spinal thermal ablation with varied modalities as generally safe when careful imaging guidance and thermal planning are used [26]. A review by Lu et al. similarly emphasizes that, with proper technique, RFA and vertebral augmentation have acceptable complication profiles in spinal metastases [33]. Pojskic et al. underscore that intraoperative monitoring, margin planning, and procedural safeguards are key to minimizing adverse events [34]. These results also reflect our findings of no significant increase in neurologic complications with RFA. One caveat is that our reliance on administrative codes to capture neurologic events cannot detect subclinical or mild sensory deficits that may not be coded, and frequency of coding may vary by institution.

### 4.4. Pain Burden with Opioid Proxy

We used prescription of morphine or oxycodone as a surrogate indicator of pain burden, and found no significant difference in utilization between groups. This suggests that, at a population level, adjunctive RFA did not lead to reduced reliance on opioids over one year. This finding is somewhat unexpected given that pain relief is one of the most consistently reported benefits of RFA in published series. Lu et al. summarize multiple reports in which RFA, which is often combined with vertebral augmentation, achieves marked reductions in pain scores and analgesic requirements in spinal metastases [33]. A recent review also reported significant pain reduction and favorable safety profiles when RFA is combined with vertebroplasty [35]. Other studies further support these findings of robust pain reductions, although their comparison was not to a matched non-RFA cohort [36,37,38,39]. The discrepancy between our results and these reports may reflect several possibilities. First, many prior studies enroll patients specifically for refractory pain and thus may preferentially include those most likely to benefit, whereas our cohort is broader and includes patients treated with RFA for other indications or in heterogeneous pain states [40]. Second, opioid prescriptions may not perfectly capture pain experience, as some patients may have pain but choose non-opioid pain management, or prescribing practices may differ across institutions [41]. Accordingly, our opioid-based proxy may be limited by formulary variability, regional prescribing differences, and the absence of non-opioid analgesic data, potentially diluting the detection of RFA’s true analgesic effect, particularly when combined with concurrent radiotherapy or kyphoplasty in a broad-matched cohort. Third, the marginal incremental analgesic benefit of RFA beyond radiotherapy and kyphoplasty may be small and diluted when assessed across a broad matched population [16]. In other words, while RFA may offer pain relief in selected patients, that benefit did not translate to a measurable difference in opioid prescribing within our study design.

### 4.5. Strengths and Limitations

Strengths of this study include the large sample size, multi-institutional scope, and use of propensity score matching to mitigate confounding. Nevertheless, limitations must be acknowledged. Coding-based data may not capture important clinical nuances such as lesion size, spinal level involvement, or patient ambulatory status. Moreover, pain burden was inferred through opioid prescriptions rather than direct patient-reported measured, which may underestimate or overestimate true symptom severity. Additionally, our follow-up window was limited to one year, potentially missing longer-term differences in disease control or late complications.

### 4.6. Future Directions

Future prospective registry and randomized studies are needed to clarify the role of RFA in metastatic spine disease, particularly in lung cancer populations. Research should focus on identifying subgroups most likely to benefit from RFA, such as those with oligometastatic disease, radioresistant histology, or tumors in mechanically critical vertebrae. Comparative studies evaluating radiotherapy combined with RFA versus radiotherapy combined with kyphoplasty may further delineate relative benefits in pain control, local tumor control, and mechanical stabilization. Incorporation of patient-reported outcomes measures including validated pain and quality-of-life metrics, as well as longer follow-up intervals, will also be critical for understanding the efficacy of RFA in this context. Advances in imaging-guided thermal ablation techniques and intra-procedural monitoring may also help refine patient selection and procedural safety.

## 5. Conclusions

In this propensity-matched real-world analysis of lung cancer spinal metastases, adding radiofrequency ablation to kyphoplasty and radiotherapy was not associated with statistically significant improvements in one-year mortality, tumor recurrence, neurologic complications, or opioid-based pain metrics. These results suggest that RFA does not consistently provide incremental benefit in this context and should be reserved for carefully selected cases. Longer follow-up beyond one year may be necessary to capture delayed differences in disease control or pain outcomes. Prospective studies incorporating patient-reported outcomes and extended follow-up are necessary to elucidate the precise role of RFA in multimodal spinal metastasis management.

## Figures and Tables

**Figure 1 healthcare-13-03101-f001:**
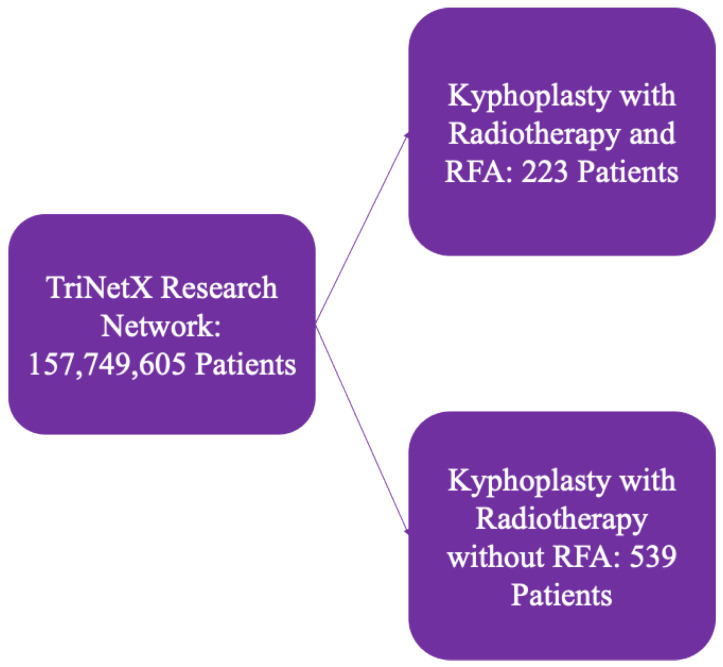
Outline of patient selection criteria from TriNetX.

**Figure 2 healthcare-13-03101-f002:**
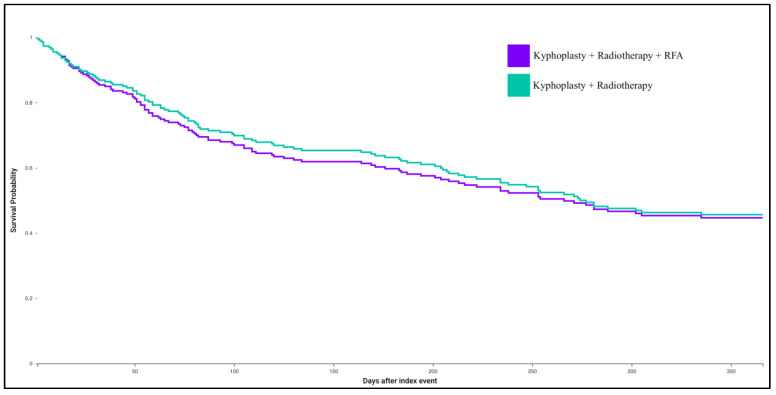
Kaplan–Meier survival curve for kyphoplasty with radiotherapy and RFA, and kyphoplasty with radiotherapy without RFA cohorts.

**Table 1 healthcare-13-03101-t001:** Patient demographics for 1-year outcomes of kyphoplasty with radiotherapy and RFA, and kyphoplasty with radiotherapy without RFA cohorts.

Cohort Demographics					
Characteristics	Unmatched Sample No. (%)		Propensity Score-Matched Sample No. (%)		
	**Kyphoplasty with Radiotherapy and RFA**	**Kyphoplasty with Radiotherapy Without RFA**	**Kyphoplasty with Radiotherapy and RFA**	**Kyphoplasty with Radiotherapy Without RFA**	**SMD**
	N = 202	N = 501	N = 202	N = 202	
Age at Index (years)	65.1	66.1	65.1	65.2	0.0041
Gender					
Female	92 (45.54%)	252 (50.30%)	92 (45.54%)	91 (45.05%)	0.0099
Male	110 (54.45%)	249 (49.70%)	110 (54.45%)	111 (54.95%)	0.0099
Race					
White	146 (72.28%)	382 (76.25%)	146 (72.28%)	147 (72.77%)	0.0111
Hispanic or Latino	10 (4.95%)	20 (3.99%)	10 (4.95%)	11 (5.45%)	0.0223
Black or African American	19 (9.41%)	42 (8.38%)	19 (9.41%)	19 (9.41%)	0
Asian	23 (11.39%)	40 (7.98%)	23 (11.39%)	22 (10.89%)	0.0157
Native Hawaiian or Pacific Islander	0	0	0	0	-
Chronic Medical Conditions					
Heart Failure	28 (13.86%)	74 (14.77%)	28 (13.86%)	27 (13.37%)	0.0144
Type 2 Diabetes Mellitus	42 (20.79%)	116 (23.15%)	42 (20.79%)	42 (20.79%)	0
Chronic Kidney Disease	29 (14.36%)	79 (15.77%)	29 (14.36%)	29 (14.36%)	0
Chronic Ischemic Heart Disease	52 (25.74%)	172 (34.33%)	52 (25.74%)	52 (25.74%)	0
Prior Treatments					
Chemotherapy Exposure	79 (39.11%)	215 (42.91%)	79 (39.11%)	78 (38.61%)	0.0102
Prior Radiation Encounter	82 (40.59%)	206 (41.12%)	82 (40.59%)	83 (41.09%)	0.0091

**Table 2 healthcare-13-03101-t002:** Risk ratios for kyphoplasty with radiotherapy and RFA, and kyphoplasty with radiotherapy without RFA cohorts.

Complications Risk						
Outcomes	Patients with Outcome in Cohort with RFA	Patients with Outcome in Cohort Without RFA	Risk Ratio	Lower Bound	Upper Bound	*p*-Value
1-year Mortality	96	94	1.021	0.83	1.256	0.8420
Neurologic Complications	18	17	1.052	0.563	1.967	0.8732
Tumor Recurrence	86	87	0.989	0.789	1.238	0.9199
Pain Burden	72	73	0.986	0.76	1.28	0.9174

## Data Availability

Restrictions apply to the availability of these data. Data were obtained from TriNetX, LLC and are available from the authors with the permission of TriNetX, LLC.

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
