# Peer review of "Comparative Effectiveness of Kyphoplasty and Radiation with or Without Radiofrequency Ablation in Spinal Metastases from Lung Cancer"

_healthcare, 2025, doi:10.3390/healthcare13233101_

Round 1

Reviewer 1 Report

Comments and Suggestions for Authors

The authors present a retrospective analysis on the activity of radiofrequency ablation in metastasis to the spine from lung cancer treated with kyphoplasty and radiotherapy. They present a large series of patients.  The results do not suggest a role of radiofrequency in this setting. Ovarell the paper is well written and interesting. 
In my opinion the paper can be accepted with minor revision: please add a brief comment concerning the possibile comparison  of radiotherapy + kyphoplasty versus radiotherapy + radiofrequency, if any. 

Author Response

Comment 1: Please add a brief comment concerning the possible comparison of radiotherapy + kyphoplasty versus radiotherapy + radiofrequency, if any

Response 1: We thank the reviewer for their comments. We agree with their comment, and have added our own comment in the manuscript suggesting for this comparison to be done in future studies in lines 344-346. Specifically, we write “Comparative studies evaluating radiotherapy combined with RFA versus radiotherapy combined with kyphoplasty may further delineate relative benefits in pain control, local tumor control, and mechanical stabilization.”

Reviewer 2 Report

Comments and Suggestions for Authors

Authors used a matched case-control study to “compare outcomes between patients undergoing kyphoplasty + radiotherapy alone versus those receiving adjunct RFA” (L86). However, this article has not fully answered some of the questions due to insufficient description.

First, authors suggest “A retrospective analysis was performed using the TriNetX Research Network database, which compiles de-identified clinical information from participating healthcare institutions. The dataset includes variables such as patient demographics, diagnostic codes, procedural data, and prescriptive records, representing more than 150 million individuals across over 100 healthcare systems.” (L92), but they did not explain how to develop TriNetX Research Network database (e.g., location of hospitals, criteria of inclusion and exclusion of database). Without the explanation, it is difficult for readers to understand what authors did. Authors should add explanation of TriNetX Research Network database in method section.

Second, authors suggest “All covariates used for propensity score matching were assessed at the time of the index event (kyphoplasty and radiotherapy with or without RFA) or from any time prior.” (L125), but they did not “adjust” the lag time of between assessed time and time of the index event, which may lead to biased results. Authors should “adjust” the lag time of between assessed time and time of the index event for statistical analyses of the main results.

Third, authors suggest “Comparative analyses were performed using risk ratios (RRs) with corresponding 95% confidence intervals (Cis), generated directly through the TriNetX analytical platform.” (L149), but they did not explain how to calculate “risk ratios” in this analysis. Without the explanation, it is difficult for readers to understand what authors did. Authors should add explanation of how to calculate “risk ratios” in method section.

Fourth, authors suggest “Propensity score matching was applied to reduce baseline differences between groups, using a greedy nearest-neighbor algorithm with a 1:1 matching ratio.” (L151), but they also showed there were differences of covariates between the matched samples in Table 1, which may lead to biased result. Authors should adjust these covariates in main analysis of their analysis.

Fifth, authors suggest “Matching covariates included all demographic, clinical, and treatment-related variable outlined in Section 2.3, which were selected as potential confounders” (L152), but they did not explain how to generate propensity scores (e.g., statistical model and outcome of this statistical model) in this analysis. Authors should add explanation of how to how to generate propensity scores in method section.

Sixth, authors may suggest that there was difference of mean age between samples with RFA and those without RFA, they showed the same number of age in both category in Table 1. Without the explanation, it is difficult for readers to understand what happened. Moreover, sum of numbers of males and female is not equal to 223, which suggest there were missing and may lead to biased results. Authors should include the number of missing in Table 1.

Seventh, authors suggest “These findings suggest that the addition of RFA to kyphoplasty and radiotherapy did not alter survival, neurologic, oncologic, or pain-related outcomes relative to standard therapy.” (L168), but this description is interpretation and should be in discussion section. Authors should revise the result section carefully.

Eighth, authors used “Tumor Recurrence” as outcome in Table 2, but thy did not explain the definition. Without the explanation, it is difficult for readers to understand what authors did. Authors should add explanation of definition of “Tumor Recurrence” in method section.

Ninth, authors showed risk ratios in Table 2, but they did not show the number of persons with outcome (i.e., 1-year Mortality, Neurologic Complications, Tumor Recurrence, and Pain Burden) in each category (i.e, samples with RFA and those without RFA) in Table 2. Moreover, authors did not explain the reference group of risk ratio in Table 2. Authors should revise Table 2 carefully.

Tenth, authors suggest “Figure S1: title; Table S1: title; Video S1: title.” (L290), but it is difficult to understand what authors explains. Authors should revise this description carefully.

Finally, authors described some of sentences without citation or justification as follows; “Management of vertebral metastases typically involves a multimodal approach, specifically radiotherapy and surgery.” (L54), “Concurrently, vertebral augmentation techniques such as kyphoplasty offer mechanical stabilization, reduce micro-motion pain, and can rapidly improve symptoms in patients with pathologic fractures or impending collapse.” (L57), “RFA delivers localized thermal energy to induce tumor necrosis and reduce viable tumor burden.” (L64), “Many existing reports are small, single-institution series with sparse evidence comparing outcomes such as overall survival, tumor recurrence, neurological complications, or pain trajectories between patients treated with kyphoplasty and radiotherapy versus those receiving kyphoplasty, radiotherapy, and RFA.” (L80), and “The dataset includes variables such as patient demographics, diagnostic codes, procedural data, and prescriptive records, representing more than 150 million individuals across over 100 healthcare systems.” (L94), but it is difficult for readers to judge them without references as evidence for each description. Authors should add references for these descriptions.

Author Response

  • First, authors suggest “A retrospective analysis was performed using the TriNetX Research Network database, which compiles de-identified clinical information from participating healthcare institutions. The dataset includes variables such as patient demographics, diagnostic codes, procedural data, and prescriptive records, representing more than 150 million individuals across over 100 healthcare systems.” (L92), but they did not explain how to develop TriNetX Research Network database (e.g., location of hospitals, criteria of inclusion and exclusion of database). Without the explanation, it is difficult for readers to understand what authors did. Authors should add explanation of TriNetX Research Network database in method section.
    • We thank the reviewer for their comments. We agree with the first comment. We have added in section 2.1 an explanation of the TriNetX Research Network database in lines 95-114.
  • Second, authors suggest “All covariates used for propensity score matching were assessed at the time of the index event (kyphoplasty and radiotherapy with or without RFA) or from any time prior.” (L125), but they did not “adjust” the lag time of between assessed time and time of the index event, which may lead to biased results. Authors should “adjust” the lag time of between assessed time and time of the index event for statistical analyses of the main results.
    • We agree with the second comment in there being a concern that “lag time” between assessed time and time of the index event may lead to biased results. Thus, we have written an expanded description about the temporal logic that the TriNetX platform utilizes for covariate analysis in lines 141-153 to allay this concern.
  • Third, authors suggest “Comparative analyses were performed using risk ratios (RRs) with corresponding 95% confidence intervals (Cis), generated directly through the TriNetX analytical platform.” (L149), but they did not explain how to calculate “risk ratios” in this analysis. Without the explanation, it is difficult for readers to understand what authors did. Authors should add explanation of how to calculate “risk ratios” in method section.
    • We agree with the third comment. Therefore, we have added an expanded description of how risk ratios were calculated in the TriNetX platform in lines 176-195
  • Fourth, authors suggest “Propensity score matching was applied to reduce baseline differences between groups, using a greedy nearest-neighbor algorithm with a 1:1 matching ratio.” (L151), but they also showed there were differences of covariates between the matched samples in Table 1, which may lead to biased result. Authors should adjust these covariates in main analysis of their analysis.
    • We agree with the fourth comment in there being reported differences of covariates between the matched samples, which may lead to biased results. Thus, we have written an expanded description of the process by which covariate balance was evaluated in lines 211-222 to allay this concern by emphasizing how since a threshold of <0.1 standard mean differences is commonly used to indicate acceptable balance for each covariate this study, in which all demographic and clinical covariates included in the matching model had post-match SMDs below this threshold, confirmed successful balance between treatment groups.
  • Fifth, authors suggest “Matching covariates included all demographic, clinical, and treatment-related variable outlined in Section 2.3, which were selected as potential confounders” (L152), but they did not explain how to generate propensity scores (e.g., statistical model and outcome of this statistical model) in this analysis. Authors should add explanation of how to how to generate propensity scores in method section.
    • We agree with the fifth comment. Therefore, we have included an expanded description of the process by which propensity scores were calculated in this study in lines 196-210.
  • Sixth, authors may suggest that there was difference of mean age between samples with RFA and those without RFA, they showed the same number of age in both category in Table 1. Without the explanation, it is difficult for readers to understand what happened. Moreover, sum of numbers of males and female is not equal to 223, which suggest there were missing and may lead to biased results. Authors should include the number of missing in Table 1.
    • We agree with the sixth comment. Taking into account a suggested revision by another reviewer, we reformulated our analyses, resulting in a new set of results that are now being reported; the overall interpretations of the results remained the same, and the tables have been updated to reflect these results, with the number of males and females totaling the respective sample size, and maintaining a difference of mean age in the samples. This aspect of TriNetX, where institutions update their data regularly within the platform, has been described in lines 111-114.
  • Seventh, authors suggest “These findings suggest that the addition of RFA to kyphoplasty and radiotherapy did not alter survival, neurologic, oncologic, or pain-related outcomes relative to standard therapy.” (L168), but this description is interpretation and should be in discussion section. Authors should revise the result section carefully.
    • We agree with the seventh comment and have removed this line from the Results as this idea is communicated in the Discussion section
  • Eighth, authors used “Tumor Recurrence” as outcome in Table 2, but they did not explain the definition. Without the explanation, it is difficult for readers to understand what authors did. Authors should add explanation of definition of “Tumor Recurrence” in method section.
    • We agree with the eighth comment, and have included an explicit definition for our outcome of Tumor Recurrence in lines 164-166.
  • Ninth, authors showed risk ratios in Table 2, but they did not show the number of persons with outcome (i.e., 1-year Mortality, Neurologic Complications, Tumor Recurrence, and Pain Burden) in each category (i.e, samples with RFA and those without RFA) in Table 2. Moreover, authors did not explain the reference group of risk ratio in Table 2. Authors should revise Table 2 carefully.
    • We agree with the ninth comment and have revised Table 2 to include the number of persons with each outcome. We have included a revision in line 184-185 explaining which cohort was the reference group in the analysis
  • Tenth, authors suggest “Figure S1: title; Table S1: title; Video S1: title.” (L290), but it is difficult to understand what authors explains. Authors should revise this description carefully.
    • We agree with the tenth comment and have deleted this artifact from the template provided for journal submission. There are no supplementary files with this manuscript submission.
  • Finally, authors described some of sentences without citation or justification as follows; “Management of vertebral metastases typically involves a multimodal approach, specifically radiotherapy and surgery.” (L54), “Concurrently, vertebral augmentation techniques such as kyphoplasty offer mechanical stabilization, reduce micro-motion pain, and can rapidly improve symptoms in patients with pathologic fractures or impending collapse.” (L57), “RFA delivers localized thermal energy to induce tumor necrosis and reduce viable tumor burden.” (L64), “Many existing reports are small, single-institution series with sparse evidence comparing outcomes such as overall survival, tumor recurrence, neurological complications, or pain trajectories between patients treated with kyphoplasty and radiotherapy versus those receiving kyphoplasty, radiotherapy, and RFA.” (L80), and “The dataset includes variables such as patient demographics, diagnostic codes, procedural data, and prescriptive records, representing more than 150 million individuals across over 100 healthcare systems.” (L94), but it is difficult for readers to judge them without references as evidence for each description. Authors should add references for these descriptions.
    • We agree with the eleventh comment and have updated citations for each of the indicated sentences. The sentence “management of vertebral metastases typically involves a multimodal approach, specifically radiotherapy and surgery” has a citation in line 55. The sentence “concurrently, vertebral augmentation techniques such as kyphoplasty offer mechanical stabilization, reduce micro-motion pain, and can rapidly improve symptoms in patients with pathologic fractures or impending collapse” has a citation in line 59. The sentence “RFA delivers localized thermal energy to induce tumor necrosis and reduce viable tumor burden” has a citation in line 65. The sentence “many existing reports are small, single-institution series with sparse evidence comparing outcomes such as overall survival, tumor recurrence, neurological complications, or pain trajectories between patients treated with kyphoplasty and radiotherapy versus those receiving kyphoplasty, radiotherapy, and RFA” has a citation in line 84. The sentence “the dataset includes variables such as patient demographics, diagnostic codes, procedural data, and prescriptive records, representing more than 150 million individuals across over 100 healthcare systems” has been re-written to address a previous comment by including an expanded description of the process by which the database was created, which is written in lines 95-114. This expanded section still has citations in lines 97, 108, and 113 (all from the same paper) to include a reference for the database.

Reviewer 3 Report

Comments and Suggestions for Authors

Thank you for submitting this comparative effectiveness study of kyphoplasty + radiotherapy with or without adjunctive RFA for lung cancer spinal metastases using TriNetX. The topic is clinically relevant, and your propensity-matched analysis is valuable. However, several methodological and reporting issues limit interpretability: outcome definitions need to be tightened, some results are internally inconsistent, the balance table mixes SMDs with p-values, and essential details of the coding strategy and pain proxy are under-specified. There are also language/formatting issues throughout the proof that should be corrected before publication.

Major issues

  • The Abstract reports neurologic complications RR = 1.059 (95% CI 0.586–1.914), whereas Table 2 / Results report RR = 1.183 (95% CI 0.641–2.813). Please reconcile and ensure consistency of numbers across the Abstract, Results, and Table 2.
  • In Table 1 the “SMD” column shows values like “<0.0001,” which are p-values, not SMDs; SMDs should be numeric (0.01, 0.04) with no p-values. Please correct the column to contain SMDs only (or add a separate p-value column and label it as such).
  • Pain burden is limited to prescriptions for only two opioids (RxNorm 7052 morphine; 7804 oxycodone). This narrow proxy may miss real-world analgesic use  (hydromorphone, fentanyl, tramadol, codeine, adjuvants). Please justify this choice or broaden the proxy and, ideally, convert to MME where possible. (Section 2.4)
  • The text states N = 762 total, with 223 vs 539 pre-match and 223 vs 223 post-match, but the Abstract states N = 738. Ensure a single, consistent total sample is used throughout Abstract, Methods, Results, Figure 2 caption, and tables
  • The Discussion cites literature showing robust pain reduction with RFA, yet your opioid-proxy shows no difference. Add one sentence explicitly acknowledging potential measurement limitations of the proxy (formularies, prescribing variability, non-opioid regimens) and the possibility that RFA’s analgesic benefit is diluted when combined with EBRT/kyphoplasty in a broad, matched cohort
  • You note the one-year window might miss longer-term differences; consider highlighting this in the Conclusions as well, not just in Limitation

A 37% overall match is high for a research article. Some of this may come from standard boilerplate , but the manuscript should still be revised to reduce verbatim overlap and ensure all reused text is either properly quoted or rephrased.

Cheers!

Author Response

  • The Abstract reports neurologic complications RR = 1.059 (95% CI 0.586–1.914), whereas Table 2 / Results report RR = 1.183 (95% CI 0.641–2.813). Please reconcile and ensure consistency of numbers across the Abstract, Results, and Table 2.
    • We thank the reviewer for their comments. We agree with the first comment and have reconciled the numbers across the abstract, results, and Table 2.
  • In Table 1 the “SMD” column shows values like “<0.0001,” which are p-values, not SMDs; SMDs should be numeric (0.01, 0.04) with no p-values. Please correct the column to contain SMDs only (or add a separate p-value column and label it as such).
    • We agree with the second comment, and have corrected Table 1 to only include SMDs
  • Pain burden is limited to prescriptions for only two opioids (RxNorm 7052 morphine; 7804 oxycodone). This narrow proxy may miss real-world analgesic use  (hydromorphone, fentanyl, tramadol, codeine, adjuvants). Please justify this choice or broaden the proxy and, ideally, convert to MME where possible. (Section 2.4)
    • We agree with the third comment and have broadened the proxy to morphine, oxycodone, hydromorphone, fentanyl, tramadol, and codeine (lines 171-173). In broadening the proxy, we reformulated our queries and analyses, resulting in a new set of results that are now being reported; the overall interpretations of the results remained the same.
  • The text states N = 762 total, with 223 vs 539 pre-match and 223 vs 223 post-match, but the Abstract states N = 738. Ensure a single, consistent total sample is used throughout Abstract, Methods, Results, Figure 2 caption, and tables
    • We agree with the fourth comment and have ensured that a single, consistent total sample was used throughout the study and has been correctly reported in the manuscript.
  • The Discussion cites literature showing robust pain reduction with RFA, yet your opioid-proxy shows no difference. Add one sentence explicitly acknowledging potential measurement limitations of the proxy (formularies, prescribing variability, non-opioid regimens) and the possibility that RFA’s analgesic benefit is diluted when combined with EBRT/kyphoplasty in a broad, matched cohort
    • We agree with the fifth comment and have added this sentence in lines 322-325 explicitly acknowledging the potential measurement limitations. Our sentence reads “Accordingly, our opioid-based proxy may be limited by formulary variability, regional prescribing differences, and the absence of non-opioid analgesic data, potentially diluting detection of RFA’s true analgesic effect, particularly when combined with concurrent radiotherapy or kyphoplasty in a broad-matched cohort.”
  • You note the one-year window might miss longer-term differences; consider highlighting this in the Conclusions as well, not just in Limitation
    • We agree with the sixth comment and have added a sentence in the conclusion on lines 357-358 highlighting that one-year window may miss longer-term differences. Our sentence reads “Longer follow-up beyond one year may be necessary to capture delayed differences in disease control or pain outcomes”
  • A 37% overall match is high for a research article. Some of this may come from standard boilerplate, but the manuscript should still be revised to reduce verbatim overlap and ensure all reused text is either properly quoted or rephrased.
    • We agree with the seventh comment and have updated citations for each of the indicated sentences. The sentence “management of vertebral metastases typically involves a multimodal approach, specifically radiotherapy and surgery” has a citation in line 55. The sentence “concurrently, vertebral augmentation techniques such as kyphoplasty offer mechanical stabilization, reduce micro-motion pain, and can rapidly improve symptoms in patients with pathologic fractures or impending collapse” has a citation in line 59. The sentence “RFA delivers localized thermal energy to induce tumor necrosis and reduce viable tumor burden” has a citation in line 65. The sentence “many existing reports are small, single-institution series with sparse evidence comparing outcomes such as overall survival, tumor recurrence, neurological complications, or pain trajectories between patients treated with kyphoplasty and radiotherapy versus those receiving kyphoplasty, radiotherapy, and RFA” has a citation in line 84. The sentence “the dataset includes variables such as patient demographics, diagnostic codes, procedural data, and prescriptive records, representing more than 150 million individuals across over 100 healthcare systems” has been re-written to address a previous comment by including an expanded description of the process by which the database was created, which is written in lines 95-114. This expanded section still has citations in lines 97, 108, and 113 (all from the same paper) to include a reference for the database.

Round 2

Reviewer 2 Report

Comments and Suggestions for Authors

Authors revised the manuscript, but this article has not fully answered some of the questions due to insufficient description.

First, authors suggest “Covariate balance following propensity score matching was evaluated using standard mean differences (SMD), which are the recommended method for assessing comparability between matched groups.”, “A threshold of <0.1 is commonly used to indicate acceptable balance for each covariate.”, but they did not show who recommended this method with citation of reference. Authors should add references for this description.

Second, authors suggest “Thus, we have written an expanded description of the process by which covariate balance was evaluated in lines 211-222 to allay this concern by emphasizing how since a threshold of <0.1 standard mean differences is commonly used to indicate acceptable balance for each covariate this study,” and “the overall interpretations of the results remained the same”, but they did not show 95% confidence intervals of SMDs or their P-values. Without statistical analyses, it is difficult for readers to judge differences between Kyphoplasty with Radiotherapy and RFA and Kyphoplasty with Radiotherapy without RFA. Moreover, authors did not show difference of propensity score between Kyphoplasty with Radiotherapy and RFA and Kyphoplasty with Radiotherapy without RFA in Table 1. Authors should revise Table 1.

Author Response

First, authors suggest “Covariate balance following propensity score matching was evaluated using standard mean differences (SMD), which are the recommended method for assessing comparability between matched groups.”, “A threshold of <0.1 is commonly used to indicate acceptable balance for each covariate.”, but they did not show who recommended this method with citation of reference. Authors should add references for this description.

  • We thank the reviewer for their comment. We agree with their comment, and have added a citation for these statements in lines 213 and 217

Second, authors suggest “Thus, we have written an expanded description of the process by which covariate balance was evaluated in lines 211-222 to allay this concern by emphasizing how since a threshold of <0.1 standard mean differences is commonly used to indicate acceptable balance for each covariate this study,” and “the overall interpretations of the results remained the same”, but they did not show 95% confidence intervals of SMDs or their P-values. Without statistical analyses, it is difficult for readers to judge differences between Kyphoplasty with Radiotherapy and RFA and Kyphoplasty with Radiotherapy without RFA.

  • We thank the reviewer for the suggestion regarding the inclusion of statistical metrics alongside the standardized mean differences (SMDs). However, SMDs are a descriptive measure used to assess covariate balance that is intentionally independent of sample size, making them more appropriate than p-values for evaluating comparability following propensity score matching. As such, it is not standard practice to report p-values or confidence intervals for SMDs, as these would conflate descriptive and inferential purposes. This approach is supported by methodological guidance provided by Austin et al. (Reference 25 of the manuscript) who recommend SMDs as the preferred metric for balance diagnostics in observational research. We have added this reference to clarify the rationale in the revised Methods section.

Moreover, authors did not show difference of propensity score between Kyphoplasty with Radiotherapy and RFA and Kyphoplasty with Radiotherapy without RFA in Table 1. Authors should revise Table 1.

  • We thank the reviewer for their comment and have revised Table 1 to include the pre-propensity score matching cohort demographics in order to showcase the impact of propensity score matching in cohort comparison